# Characterization and Classification of Spatial White Matter Tract Alteration Patterns in Glioma Patients Using Magnetic Resonance Tractography: A Systematic Review and Meta-Analysis

**DOI:** 10.3390/cancers15143631

**Published:** 2023-07-15

**Authors:** Arash L. Mahmoodi, Maud J. F. Landers, Geert-Jan M. Rutten, H. Bart Brouwers

**Affiliations:** Department of Neurosurgery, Elisabeth-TweeSteden Hospital, Hilvarenbeekseweg 60, 5022 GC Tilburg, The Netherlands

**Keywords:** magnetic resonance tractography, glioma, white matter tract, spatial alteration patterns, classification systems

## Abstract

**Simple Summary:**

Tractography is a neuroimaging technique to visualize the white matter (WM) tracts in vivo. Gliomas, the most common primary brain tumor in adults, can alter the surrounding WM tracts according to various spatial patterns that can be studied using tractography. We systematically reviewed the homogeneity between the reported classification systems of these patterns in the literature, and investigated whether low-grade gliomas (LGGs) and high-grade gliomas (HGGs) preferred certain spatial WM tract alteration patterns. Four types of spatial WM tract alteration patterns were reported in the current literature: displacement, infiltration, disruption/destruction and edematous. We found a considerable heterogeneity between the reported classification systems. In a subset of studies, we found sufficient homogeneity in the classification systems to further analyze the displacement and infiltration patterns. We found that LGGs displaced WM tracts more often than HGGs, while for WM tract infiltration, no differences between LGGs and HGGs were found.

**Abstract:**

Introduction: Magnetic resonance (MR) tractography can be used to study the spatial relations between gliomas and white matter (WM) tracts. Various spatial patterns of WM tract alterations have been described in the literature. We reviewed classification systems of these patterns, and investigated whether low-grade gliomas (LGGs) and high-grade gliomas (HGGs) demonstrate distinct spatial WM tract alteration patterns. Methods: We conducted a systematic review and meta-analysis to summarize the evidence regarding MR tractography studies that investigated spatial WM tract alteration patterns in glioma patients. Results: Eleven studies were included. Overall, four spatial WM tract alteration patterns were reported in the current literature: displacement, infiltration, disruption/destruction and edematous. There was a considerable heterogeneity in the operational definitions of these terms. In a subset of studies, sufficient homogeneity in the classification systems was found to analyze pooled results for the displacement and infiltration patterns. Our meta-analyses suggested that LGGs displaced WM tracts significantly more often than HGGs (*n* = 259 patients, RR: 1.79, 95% CI [1.14, 2.79], I^2^ = 51%). No significant differences between LGGs and HGGs were found for WM tract infiltration (*n* = 196 patients, RR: 1.19, 95% CI [0.95, 1.50], I^2^ = 4%). Conclusions: The low number of included studies and their considerable methodological heterogeneity emphasize the need for a more uniform classification system to study spatial WM tract alteration patterns using MR tractography. This review provides a first step towards such a classification system, by showing that the current literature is inconclusive and that the ability of fractional anisotropy (FA) to define spatial WM tract alteration patterns should be critically evaluated. We found variations in spatial WM tract alteration patterns between LGGs and HGGs, when specifically examining displacement and infiltration in a subset of the included studies.

## 1. Introduction

Over the past few decades, the use of magnetic resonance (MR) tractography in neuroscience has greatly expanded our current anatomical knowledge of white matter (WM) tracts, illustrated by discoveries of new distinct pathways and a better understanding of existing ones [1,2,3,4,5,6]. As of today, MR tractography remains the only neuroimaging technique to non-invasively visualize subcortical WM tracts in vivo. Gliomas, the most common primary brain tumor in adults, can spread diffusely throughout the brain and WM tracts and infiltrate the surrounding cortical and subcortical tissue, complicating safe tumor resection [7,8]. MR tractography allows us to study the spatial relations between brain tumors and surrounding WM tracts and is therefore also recognized for its potential use as a tool in the neurosurgical treatment of gliomas [9].

The clinical application of MR tractography in glioma treatment has subsequently been thoroughly studied, resulting in increasing evidence supporting its use in planning surgical approaches and enhancing the extent of tumor resection (EOR) [10,11,12,13,14,15,16,17,18,19,20]. The ability to safely maximize the EOR is a main goal in glioma surgery, as a higher EOR has repeatedly been associated with prolonged survival rates [21,22,23]. Altogether, glioma resections encompass an onco-functional balance between maximizing the EOR in order to prolong survival, while maintaining surrounding cortical and subcortical areas to preserve function [17,24,25,26]. Pre- and intraoperative knowledge of the spatial relations between the tumor and the surrounding WM tracts enables neurosurgeons to tailor the resection to the unique characteristics of each patient’s tumor. These spatial relations might differ depending on tumor grade or type. Therefore, accurate visualization of the spatial relations between tumors and surrounding WM tracts, using MR tractography, could further optimize the onco-functional balance in glioma treatment [9]. 

Some of the initial studies have introduced classification systems to classify spatial WM tract alteration patterns in gliomas [27,28,29]. They differentiated between tumor displacement of WM tracts, tumor infiltration of WM tracts, disruption of WM tracts and edematous (affected by vasogenic edema) WM tracts. Both visual assessment and quantification of the WM tract’s microstructural integrity, as expressed, for example, by fractional anisotropy (FA), were used to characterize these patterns. Whereas some of the later studies continued to use these classification systems, others have made modifications. Therefore, it is unknown whether the definitions of the different spatial WM tract alteration patterns that are used in the existing literature are sufficiently similar to perform meta-analyses. Sufficient homogeneity in these definitions would allow for a conjoint analysis of results across studies, to evaluate, for example, the relation between glioma grade and specific WM tract patterns. In this review, we investigate to what extent the characterization and classification of spatial WM tract alteration patterns is uniformly reported in the current literature. Subsequently, we evaluate the possible relation between glioma grade and specific spatial WM tract alteration patterns.

## 2. Materials and Methods

### 2.1. Literature Search and Study Selection

A systematic literature search following the PRISMA guidelines was conducted to identify studies that described spatial patterns of WM tract alterations using MR tractography in glioma patients. Search terms were established and search strategies (Appendix A) were built and verified by two librarians of the medical library of the Elisabeth-TweeSteden Hospital, Tilburg, The Netherlands. The databases PubMed Central, Web of Science, Embase and Cochrane Library were searched by the first author on 4 April 2023. Studies that met the following criteria were included: studies that (1) investigated glioma patients, (2) performed pre-treatment MR tractography, (3) described spatial WM tract alteration patterns in relation to glioma grade in qualitative terms, (4) were written in English. The following types of articles were excluded: conference abstracts, single case reports, reviews, book chapters or protocols. Studies were initially screened for inclusion based on title and abstract by two authors (A.L.M. and M.J.F.L). Full text articles were read if the study eligibility could not be determined based on title and abstract screening. Included articles were reviewed by two authors (A.L.M. and M.J.F.L), and differences were adjudicated by consensus. Reviews and reference lists of included articles were screened to identify potential additional articles.

### 2.2. Data Extraction

Relevant data were manually extracted from the included studies. The following data were collected: (1) first author and year of publication, (2) sample sizes (grouped by tumor grade if possible), (3) tractography method, (4) WM tract pattern classification system, (5) proportions of WM tract patterns in relation to glioma grade. Three studies did not directly report the outcome measure of interest in their results section, i.e., the prevalence of spatial WM tract alteration patterns per glioma grade [30,31,32]. For these studies, raw data were extracted either from tables with study characteristics, or from supplementary documents. The outcome measure of interest was subsequently manually calculated from the extracted raw data.

### 2.3. Data Analysis

To evaluate the relation between glioma grade and specific spatial WM tract alteration patterns, the proportions of the displacement and infiltration patterns were compared in meta-analyses between low-grade gliomas (LGGs) and high-grade gliomas (HGGs), as we found sufficient homogeneity in the classification systems of a subset of studies to pool results. The meta-analyses were performed using Review Manager software, in which the pooled outcome measures among the included studies were statistically analyzed [33]. The outcome measures were dichotomized, and risk ratios (RRs) and 95% confidence intervals (CIs) were calculated using the Mantel–Haenszel method under a random-effects model, to account for interstudy variability. Forest plots were generated, by incorporating studies with comparable definitions of spatial WM tract alteration patterns. Studies were deemed comparable if they used similar criteria for the visual assessment of the spatial WM tract alteration patterns. The DerSimonian and Laird method was used as an interstudy variance estimator. Heterogeneity between studies was assessed using the Q test and the I^2^ statistic. Pooled outcome measures were considered statistically significant at *p* < 0.05.

### 2.4. Methodological Quality Assessment

The methodological quality of the eligible studies was assessed using the methodological index for non-randomized studies (MINORS) [34]. The MINORS tool is designed specifically to assess the methodological quality in surgical research based on non-randomized studies. Studies were scored on eight subjects using a scale of 0–2 points. In the case of a comparative study, four additional subjects were scored. Consensus was reached for every article.

## 3. Results

### 3.1. Study Selection

Our previously described literature search resulted in 1572 articles (Figure 1). We found 579 duplicates, resulting in 993 unique articles. Titles and abstracts were screened, after which 957 records were excluded. Of the remaining 36 papers, 5 studies could not be retrieved. Eligibility of the 31 retrieved reports was assessed, during which 18 articles were excluded for the following reasons: 11 articles did not describe spatial WM tract alteration patterns in relation to glioma grade, 4 articles were not of the publication type of interest, 1 study was not performed in the population of interest, and 2 full text articles were not available in English. Screening of reviews and reference lists of included studies did not identify additional articles. Therefore, 13 studies were included in this review. Ultimately, 2 out of the 13 included studies were deemed unsuitable. One of these studies reported a discrepancy between the sample sizes in the methods section and the results section [35]. We contacted the authors but we were unable to reach them for further clarification. The other study reported glioma grades in an ambiguous manner, from which it was not possible to make a clear distinction between LGG and HGG [36]. The remaining 11 suitable studies were included in this review and consisted of: 3 retrospective studies, 3 prospective studies, 2 comparative studies, 1 unspecified study design, 1 pilot study, 1 case series. The included studies were published between 2005 and 2022. The characteristics and results of the included studies are presented in Table 1.

### 3.2. Uniformity of Classification Systems for Spatial White Matter Tract Alteration Patterns

Most of the included studies combined both qualitative visual assessment and quantitative microstructural integrity of WM tracts in their definitions of spatial WM tract alteration patterns. The previously developed classification systems (Table 2) by Field et al. [27], Jellison et al. [28] and Witwer et al. [29] were used by six of the included studies [32,37,40,41,42,43]. Additionally, five studies proposed their own classification systems [30,31,38,39,44]. Of these, three relied exclusively on visual assessment of tractography results to define spatial WM tract alteration patterns, without considering quantitative FA measurements [30,39,44]. Moreover, several studies included combinations of the four commonly reported spatial WM tract alteration patterns (displacement, infiltration, disruption, and edematous), suggesting that these patterns are not mutually exclusive [30,38,39,41].

### 3.3. Relation between Glioma Grade and Spatial WM Tract Alteration Patterns

#### 3.3.1. Studies Using Jellison/Field Classification System 

Studies that used the classification systems provided by Jellison et al. [28] or Field et al. [27] are discussed together, as these systems are highly similar. Dubey et al. and Shalan et al. classified WM tract patterns according to these systems (Table 1) [41,42]. Dubey et al. found that LGGs displaced (75%, 9/12) WM tracts more often compared to HGGs (29%, 5/17). Infiltration and disruption of WM tracts were reported collectively, and occurred more in HGGs (71%, 12/17) compared to LGGs (25%, 3/12). Statistical significance was not assessed. Similar results were reported by Shalan et al., who found that displacement occurred more often in LGGs (83%, 5/6) compared to HGGs (71%, 10/14). HGGs showed more infiltration (79%, 11/14) compared to LGGs (17%, 1/6). Statistical significance was reached only for the infiltration pattern (*p* = 0.018).

#### 3.3.2. Studies Using Witwer Classification System

Four studies [32,37,40,43] used the Witwer et al. [29] classification system (Table 1). Bakhshi et al. reported higher frequencies of displacement (30%, 14/43) and infiltration (63%, 27/43) in HGGs, compared to the frequencies of displacement (25%, 5/20) and infiltration (55%, 11/20) in LGGs. Overall, infiltration occurred most frequently, irrespective of tumor grade. Gao et al. found more displacement in LGGs (92%, 12/13) compared to HGGs (28%, 9/32). Infiltration was found more often in HGGs (16%, 5/32) compared to LGGs (8%, 1/13). Zhang et al. reported higher frequencies of displacement in LGGs (100%, 2/2) compared to HGGs (0%, 0/7). Infiltration was observed in 100% of the cases for both LGGs and HGGs. Deilami et al. investigated multiple WM tracts per patient, due to which the number of studied WM tracts was not equal to the number of included patients. Displacement in LGGs (31%) was observed more frequently compared to HGGs (20%). Infiltration was also observed more frequently in LGGs (65%) compared to HGGs (49%).

#### 3.3.3. Studies Using Self-Described Classification Systems

Five studies [30,31,38,39,44] used self-described classification systems to classify spatial WM tract alteration patterns (Table 1). It should be noted that precise definitions of similarly named patterns (e.g., “displacement”) could therefore vary between these studies. Camins et al. adapted the classification system of Witwer et al. by combining the edematous and infiltration patterns. Displacement was found more frequently in LGGs (33%, 2/6) compared to HGGs (11%, 3/28). Combined infiltrated and edematous tracts were found more frequently in HGGs (64%, 18/28) compared to LGGs (50%, 3/6). However, their observed results were not significantly different between LGGs and HGGs. Celtikci et al. evaluated only grade II gliomas, and analyzed multiple WM tracts per patient. Based on the analysis of 65 tracts in 16 patients, displacement (37%), infiltration (20%) and combined displacement and infiltration (29%) were all reported. Delgado et al. found an equal propensity for WM tract infiltration (*p* = 0.75) between LGGs (77%, 17/22) and HGGs (92%, 11/12). Displacement was reported more frequently in LGGs (55%, 12/22) compared to HGGs (42%, 5/12), but no statistics were performed for this pattern. Yu et al. analyzed multiple tracts per patient and differentiated between isolated displacement, isolated disruption and combined displacement and disruption. In LGGs, isolated displacement (17%) was observed more frequently compared to HGGs (7%). Combined displacement and disruption was observed in HGGs (30%), while this was not observed in LGGs. Zhukov et al. reported displacement more frequently in LGGs (30%, 3/10) compared to HGGs (20%, 3/15). Infiltration was reported more frequently in HGGs (40%, 6/15) compared to LGGs (20%, 2/10). 

### 3.4. Meta-analysis

Three studies were excluded prior to performing the meta-analysis. Celtikci et al. only included LGGs in their study and could therefore not be compared to HGGs [39]. Deilami et al. and Yu et al. reported the occurrence of displacement across multiple tracts per patient, which artificially increased the sample size and precluded meaningful statistical comparisons to the other included studies [30,40]. For the analysis of the infiltration pattern, two more studies from Camins et al. and Dubey et al. were excluded, since they evaluated combined patterns that consisted of infiltration and other patterns, resulting in pooled results that did not solely represent the infiltration pattern [38,41].

#### 3.4.1. Displacement of WM Tracts in LGGs versus HGGs

The results of eight studies were deemed suitable to evaluate the pooled relation between glioma grade and WM tract displacement (Figure 2) [31,32,37,38,41,42,43,44]. In total, displacement of WM tracts was reported in 50 out of 91 (54.9%) LGG cases, and in 48 out of 168 (28.6%) HGG cases. The forest plot containing the pooled result (Figure 2) indicates that WM tract displacement occurred significantly more often in LGGs compared to HGGs (RR: 1.79, 95% CI [1.14, 2.79], *p* = 0.01, I^2^ = 51%).

#### 3.4.2. Infiltration of WM Tracts in LGGs versus HGGs

The results of six studies were deemed suitable to evaluate the pooled relation between glioma grade and WM tract infiltration (Figure 3) [31,32,37,42,43,44]. In total, infiltration of WM tracts was reported in 34 out of 73 (46.6%) LGG cases, and in 67 out of 123 (54.5%) HGG cases. The forest plot containing the pooled result (Figure 3) indicated that the proportion of WM tract infiltration did not vary significantly between LGGs and HGGs (RR: 1.19, 95% CI [0.95, 1.50], *p* = 0.13, I^2^ = 4%).

### 3.5. Methodological Quality Assessment

The mean MINORS score (Appendix A) for the nine non-comparative studies was 9.3 ± 1.87 (range 7–12). The mean score for the two comparative studies was 16.5 ± 2.12. One study was a case series and could therefore not be scored on the following subjects: inclusion of consecutive patients, loss to follow up less than 5% and prospective calculation of the study size. Only two studies reported on the unbiased assessment of the study endpoint. None of the studies reported on the prospective calculation of the study size.

## 4. Discussion

Gliomas can alter the surrounding WM tracts according to various spatial patterns that can be studied using MR tractography. To our knowledge, this is the first systematic review and meta-analysis on spatial WM tract alteration patterns in glioma patients. We investigated to what extent the characterization and classification of spatial WM tract alteration patterns is uniformly reported in the current literature, and we subsequently evaluated the relation between glioma grade and these patterns. In general, four spatial patterns were reported: displacement, infiltration, disruption/destruction and edematous. A considerable heterogeneity in the definitions of these patterns was found among the reported classification systems. Nonetheless, in a subset of studies, sufficient homogeneity was found to analyze pooled results for the displacement and infiltration patterns. Our meta-analyses of this subset of studies suggested that LGGs were more likely to displace WM tracts as compared to HGGs. LGGs and HGGs were equally probable to infiltrate WM tracts in this study. However, reliable interpretations of these pooled results were hampered by the multifactorial heterogeneity that was present in our study.

We were unable to perform meta-analyses on the edematous and disruption/destruction patterns in the current study. With regard to the edematous pattern, not enough studies reported on this. With regard to the disruption/destruction pattern, studies did not report if WM tracts were actually destroyed by the tumor, or if the tumor disturbed the brain’s anatomy beyond the point at which the tractography algorithm could no longer identify WM tracts. 

The heterogeneity of WM tract pattern definitions found among classification systems is largely due to the use of microstructural integrity measurements, i.e., FA, by the majority of the included studies. Previous studies that aimed to differentiate between spatial WM tract alteration patterns using FA have shown that FA is not uniformly affected by particular patterns [36,40]. Deilami et al. found an increased FA to be associated with WM tract displacement, while a decreased FA was associated with either infiltration or disruption. Yen et al. found a decreased FA in WM tract disruption, while edema, displacement or infiltration did not affect the microstructural integrity of WM tracts. Possible drawbacks (due to crossing fibers) of using FA as a biomarker for the microstructural integrity of WM tracts are illustrated by Figley et al. [45]. Another factor that could account for heterogeneity between studies is that MR tractography is considered a user-dependent technique. Automated tractography protocols are currently being developed, which could further reduce human error and interobserver variation in future studies [46,47]. It is also argued that diffusion tensor imaging (DTI) tractography, the tractography method used by all but one of the included studies in this review, is not the most accurate method to investigate spatial WM tract alteration patterns. Some have advocated that DTI tractography was suboptimal to discriminate between WM tracts surrounded by edema or tumor infiltrated tracts, while others have contradicted this statement [38,48,49]. Novel tractography methods, such as constrained spherical deconvolution (CSD), are technically better equipped to discriminate between crossing fibers within a single voxel as compared to DTI tractography [50]. Altogether, the question of whether spatial WM tract alteration patterns should be characterized based on quantitative microstructural integrity measurements remains to be evaluated. More in general, the aforementioned shortcomings are all inherent to the study design of a meta-analysis based on observational data. However, although the observed I^2^ might implicate a feasible homogeneity for the performed meta-analyses, the overall pooled results should be interpreted with caution, given the large and multifactorial inter-study heterogeneity that was present.

A limitation of our review is that we were unable to identify any cut-off distances between tumors and WM tracts beyond which patients would be excluded from the included studies. WM tracts located distant from a tumor might sustain different, and probably less noticeable, spatial alterations as compared to tracts in close proximity to a tumor. This could have resulted in both under- or overestimations of the reported findings. Another limitation that has introduced additional heterogeneity and hampered the interpretations of our meta-analyses is the use of different World Health Organization (WHO) tumor classification systems among the included studies. Not every study reported on this, and studies that did report the WHO classification system either used the 2007 or updated 2016 version. We assigned patients to LGG or HGG groups as defined in their respective studies. The 2016 WHO classification system integrated molecular markers to classify tumors, as compared to the 2007 WHO classification system that was exclusively based on histology. Therefore, patients could not always be consistently assigned to LGG or HGG groups in our study, which could have negatively affected the uniformity of our pooled results.

To conclude, our findings emphasize the need for a clear, consensus-based classification system for spatial WM tract alteration patterns in gliomas, to allow for more accurate comparisons of results across different studies. Therefore, future research should focus on the development of an objective and reproducible classification system of spatial WM tract alteration patterns. This review provides a first step towards such a classification system, by showing that the current literature is inconclusive and that the ability of FA to define spatial WM tract alteration patterns should be critically evaluated in larger prospective studies in a controlled setting.

## 5. Conclusions

Researchers and clinicians have expressed great interest in elucidating the possible spatial WM tract alteration patterns in glioma patients, using MR tractography. In this review, we found a variety of reported patterns, but the interpretation of results from different studies was complicated by a lack of a uniform classification system and methodological heterogeneity. Our meta-analyses suggested that LGGs were more likely to displace WM tracts as compared to HGGs, while for WM tract infiltration, no significant differences between LGGs and HGGs were found. The limited evidence highlights the need for a consensus-based classification system for spatial WM tract alteration patterns, to better enable comparisons between studies. A uniform classification system would further establish MR tractography as a non-invasive tool to study the spatial relations of gliomas with the surrounding WM tracts, ultimately advancing the field of neurosurgery.

## Figures and Tables

**Figure 1 cancers-15-03631-f001:**
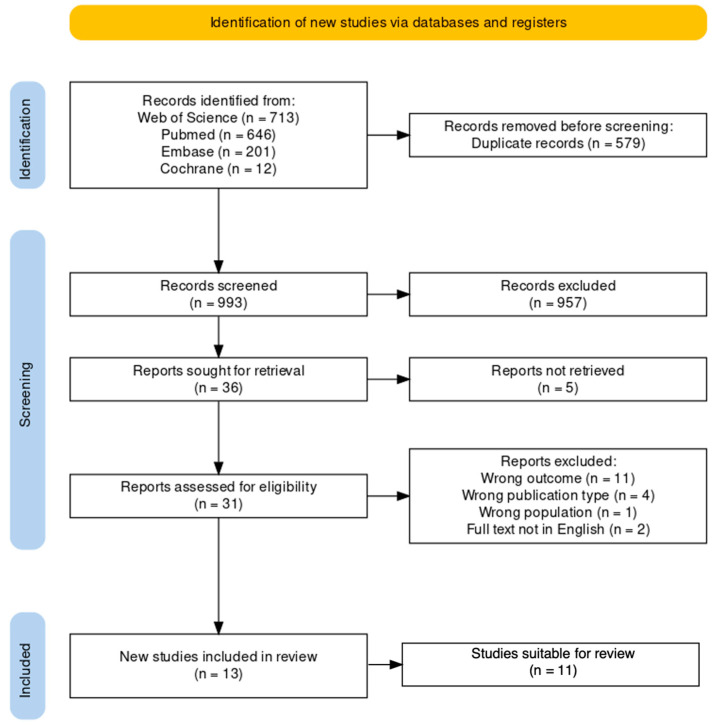
PRISMA flow diagram displaying the process of article identification and inclusion.

**Figure 2 cancers-15-03631-f002:**
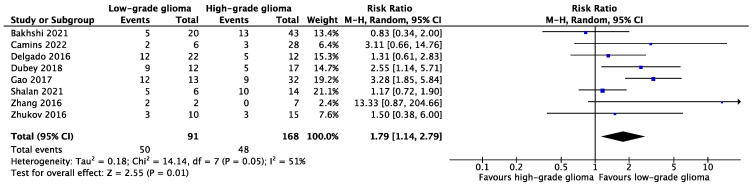
Forest plot of comparison of WM tract displacement between LGGs and HGGs [31,32,37,38,41,42,43,44].

**Figure 3 cancers-15-03631-f003:**
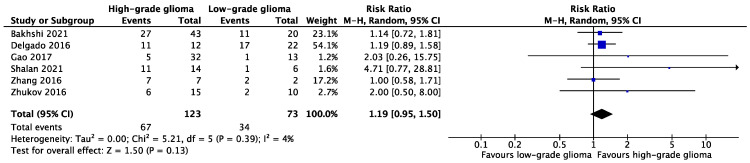
Forest plot of comparison of WM tract infiltration between LGGs and HGGs [31,32,37,42,43,44].

**Table 1 cancers-15-03631-t001:** Characteristics and results of studies that reported on spatial white matter tract alteration patterns in glioma patients.

Study	No. of Patients (Total)	No. of Patients (Grouped by Tumor Grade)	LGG (%)	HGG (%)	Tractography Method	WM Tract Pattern Classification System	Relation between Glioma Grade and Spatial WM Tract Alteration Patterns
Bakhshi et al. 2021 [37]	57	Astrocytoma: grade I (*n* = 3); grade II (*n* = 1); grade III (*n* = 2); GBM (*n* = 26) Oligodendroglioma: grade II (*n* = 13); grade III (*n* = 12) Oligoastrocytoma: grade II (*n* = 3); grade III (*n* = 3)	30	70	DTI tractography, fiber tracking protocol not specified	According to Witwer et al. (2002) [29]	LGG (*n* = 20): 25% displaced; 55% infiltrated; 15% disrupted HGG (*n* = 43): 30% displaced; 63% infiltrated; 7% disrupted
Camins et al. 2022 [38]	34	Grade I ganglioglioma (*n* = 1); grade I pleomorphic xanthoastrocytoma (*n* = 1) Astrocytoma: grade II diffuse astrocytoma (*n* = 4); grade III anaplastic astrocytoma (*n* = 9); GBM (*n* = 14) Oligodendroglioma: grade III anaplastic oligodendroglioma (*n* = 5)	18	82	DTI tractography using manual deterministic fiber tracking algorithm, ROI selection method described. Tractography performed by consensus by two neuroradiologists with subspecialized experience in presurgical DTI tractography mapping	Self-described classification, partly based on Jellison et al. (2004) [28] and Field et al. (2004) [27]: Intact tract; Displaced (preserved or elevated FA); Infiltrated/edematous (reduced FA and preserved tract direction, with or without displacement); Destroyed (extremely reduced FA and loss of directional data)	LGG (*n* = 6): 33% displaced; 50% combined infiltrated/edematous; 0% destroyed HGG (*n* = 28): 11% displaced; 64% combined infiltrated/edematous; 25% destroyed
Celtikci et al. 2018 [39]	16	Only grade II glioma: oligodendroglioma (*n* = 7); diffuse astrocytoma (*n* = 5); gemistocytic astrocytoma (*n* = 2); pleomorphic xanthoastrocytoma (*n* = 1); pilomyxoid astrocytoma (*n* = 1)	100	0	Fiber tracking using QA-based generalized deterministic algorithm. Qualitative evaluation by the consensus of first two authors, blinded to all clinical information	Self-described classification: Unaffected; Displaced (changed tract trajectory due to mass effect); Infiltrated (segment or entire peritumoral tract runs through tumor); Displaced and infiltrated simultaneously; Disruption (tract partially or completely interrupted)	65 tracts analyzed in 16 patients (all grade II): 14% unaffected; 37% displaced; 20% infiltrated; 29% displaced & infiltrated; 0% disrupted
Deilami et al. 2015 [40]	11	Not reported	36	64	DTI tractography, fiber tracking protocol not specified. ROI selection by a hand-drawn procedure, using an atlas for more accurate ROI drawing	According to Witwer et al. (2002) [29]	LGG (23 tracts analyzed in 4 patients): 65% infiltrated; 31% displaced; 4% disrupted; 0% edematous HGG (74 tracts analyzed in 7 patients): 49% infiltrated; 20% displaced; 19% disrupted; 12% edematous
Delgado et al. 2016 * [31]	34	Only grade II/III glioma (*n* = 34) Astrocytoma: grade II (*n* = 9); grade III (*n* = 9) Oligodendroglioma: grade II (*n* = 13); grade III (*n* = 3)	65	35	DTI tractography using TrackVis software, ROI selection based on white matter atlas	Self-described classification: Dislocated (tract deviation from expected FA-color map trajectory and located outside tumor area, defined as increased T2-FLAIR signal intensity); Infiltrated (tract runs through increased T2-FLAIR tumor area).	LGG (*n* = 22): 55% dislocated; 77% infiltrated HGG (*n* = 12): 42% dislocated; 92% infiltrated
Dubey et al. 2018 [41]	29	Not reported	41	59	DTI tractography using deterministic tracking algorithm. Data were reviewed by the senior faculty of neurosurgery	According to Jellison et al. (2004) [28]	LGG (*n* = 12): 75% displaced; 25% infiltrated/disrupted HGG (*n* = 17): 29% displaced; 71% infiltrated/disrupted
Gao et al. 2017 * [32]	45	Astrocytoma: grade I (*n* = 5); grade II (*n* = 3); grade III (*n* = 15); GBM (*n* = 15) Oligodendroglioma: grade II (*n* = 5); grade III (*n* = 1) Gliosarcoma (*n* = 1)	29	71	DTI tractography, fiber tracking using line propagation technique. ROI selection method described	Combination of Witwer et al. (2002) [29] and Field et al. (2004) [27]	LGG (*n* = 13): 92% displaced; 8% infiltrated; 0% disruption HGG (*n* = 32): 28% displaced; 16% infiltrated; 56% disrupted
Shalan et al. 2021 [42]	20	Not reported	30	70	DTI tractography, fiber tracking protocol not specified. ROI selection method described per tract	According to Jellison et al. (2004) [28]	LGG (*n* = 6): 33% unaffected; 83% displaced; 67% edematous; 17% infiltrated; 0% destructed HGG (*n* = 14) 0% unaffected; 71% displaced; 50% edematous; 79% infiltrated; 29% destructed. Significant difference for infiltration between the 2 groups (*p* = 0.018)
Yu et al. 2005 * [30]	12	Astrocytoma: grade II (*n* = 1) grade III (*n* = 2) Oligodendroglioma: grade II (*n* = 1) Oligoastrocytoma grade III (*n* = 4) GBM (*n* = 4)	17	83	DTI tractography, using probabilistic fiber tracking in anterograde and retrograde direction. ROI selection method described per tract	Self-described classification: Simple displacement (tract location altered but integrity preserved); Displacement with disruption (reduced fibers with displacement of residual tract); Simple disruption (reduced fibers without displacement of residual tract)	LGG (6 tracts analyzed in 2 patients): 17% simple displacement; 33% simple disruption; 50% not measurably involved HGG (30 tracts analyzed in 10 patients): 7% simple displacement; 7% simple disruption; 30% displacement with disruption; 57% not measurably involved
Zhang et al. 2012 [43]	9	Grade II (*n* = 2); grade III/IV (*n* = 7)	22	78	DTI tractography, fiber tracking using line propagation technique. ROI selection method described per tract	According to Witwer et al. (2002) [29]	LGG (*n* = 2): 100% displaced; 100% infiltrated; 0% disrupted HGG (*n* = 7): 0% displaced; 100% infiltrated; 100% disrupted
Zhukov et al. 2016 [44]	25	Grade I (*n* = 2); grade II (*n* = 8); grade III (*n* = 4); grade IV (*n* = 11)	40	60	DTI tractography, fiber tracking protocol and ROI selection method not specified	Self-described classification: Intact (tract position far from tumor and edema, with unchanged trajectory and tract thickness); Displaced (tract trajectory is changed and runs along tumor border); Infiltrated (tract located inside tumor, with thinner tract)	LGG (*n* = 10): 50% intact; 30% displaced; 20% infiltrated. HGG (*n* = 15): 40% intact; 20% displaced; 40% infiltrated

***** Outcomes of interest manually calculated from extracted raw data.

**Table 2 cancers-15-03631-t002:** Commonly reported reference classification systems for spatial WM tract alteration patterns.

Study	WM Tract Pattern	Definition
Witwer et al. 2002 [29]	Displacement	Normal FA maintained relative to contralateral hemisphere corresponding tract, but tract has abnormal location or abnormal orientation on color-coded orientation map.
	Infiltration	Reduced FA, but tract remains identifiable on color-coded orientation map.
	Disruption	Significantly reduced FA, and tract not identifiable on color-coded orientation map.
	Edematous	Normal FA, and orientation on color-coded orientation map maintained, but tract shows high T2-weighted signal intensity.
Jellison et al. 2004 [28]	Displacement	Normal/slightly decreased FA with abnormal location/direction on directional color maps due to bulk mass displacement.
	Infiltration	Substantially decreased FA, and abnormal hues on directional color map.
	Disruption	(Near-)isotropic FA, and tract not identifiable on directional color map.
	Edematous	Substantially decreased FA with normal location/direction on directional color map.
Field et al. 2004 [27]	Displacement	Normal/slightly decreased FA (<25%), and normal/slightly increased apparent diffusion coefficient (ADC) (<25%) relative to contralateral hemisphere, and abnormal location/direction due to bulk mass displacement.
	Infiltration	Substantially decreased FA and increased ADC, and abnormal hues on directional color map, not due to bulk mass displacement.
	Disruption	(Near-)isotropic FA, and tract not identifiable on directional color map.
	Edematous	Substantially decreased FA with increased ADC, and normal location and direction on directional color map.

Note: Definitions of WM tract patterns are quoted from the respective articles.

## Data Availability

No new data were created or analyzed in this study. Data sharing is not applicable to this article.

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
