# Peer review of "Characterization and Classification of Spatial White Matter Tract Alteration Patterns in Glioma Patients Using Magnetic Resonance Tractography: A Systematic Review and Meta-Analysis"

_cancers, 2023, doi:10.3390/cancers15143631_

Round 1

Reviewer 1 Report

This review tried to analyze reported tractgraphy changes in the patients with glioma. Because the definitions of infiltration, disruption and edematous are different for each study, the conclusion of this meta-analysis are not reliable. 

Author analyzed the relation between tumor malignancy grade and tractography. Tumor malignancy grades are come from old pathological classification. The location of tumor is not analyzed in this review.

Reviewer 2 Report

This is a good meta-analysis but is based on a small number of  heterogenous publications using different classification systems and various MRI parameters. This problem is highlighted in the conlusion.

The study can be published.

Reviewer 3 Report

This paper is written clearly and completely.  However, there are more possiblities to arise brain tumor by circulation issue, such as the willis circle biofluid mechanics.  It would be much better to increase the factors and the review article will be perfect.

This paper is written clearly and completely.  However, there are more possiblities to arise brain tumor by circulation issue, such as the willis circle biofluid mechanics.  It would be much better to increase the factors and the review article will be perfect.

Round 2

Reviewer 1 Report

The limitation of this review should be fully discussed.

What is the clinically useful lessens we learned from this review?

Author should make this clear.

Author Response

Reviewer’s response:

The limitation of this review should be fully discussed.

What is the clinically useful lessens we learned from this review?

Author should make this clear.

Authors’ response:
We would like to thank the reviewer again for their careful reading of our manuscript and their remarks. Below we present our point-by-point response to the reviewer’s comments.

Point 1:
The limitation of this review should be fully discussed.

Response 1:
We have made major additional changes to the manuscript, in order to fully discuss and clarify the limitation of our review. We discuss the multifactorial heterogeneity that is present in our study, and that the interpretation of our meta-analyses should therefore be in the context of our limitations.

Point 2:
What is the clinically useful lessens we learned from this review?

Response 2:
Enhancing our understanding of the spatial properties of gliomas can aid in the decision-making process of performing awake or asleep surgery and in determining the extensiveness of surgery. This information could also contribute to better patient counseling. Our study provides some initial results on this subject. Given the limitations of our study, further research is needed to accomplish this clinical goal.

Round 3

Reviewer 1 Report

I recognized significant improvements of this manuscript.